# Scientific Validation of Using Active Constituent as Research Focus in Traditional Chinese Medicine: Case Study of *Pueraria lobata* Intervention in Type 2 Diabetes

**DOI:** 10.3390/ph17121675

**Published:** 2024-12-12

**Authors:** Yaping Chen, Qiuqi Wen, Meng Lin, Bing Yang, Liang Feng, Xiaobin Jia

**Affiliations:** School of Traditional Chinese Pharmacy, Innovation Center for Industry-Education Integration of Pediatrics and Traditional Chinese Medicine, State Key Laboratory of Natural Medicines, China Pharmaceutical University, Nanjing 211198, China; 13382059050@163.com (Y.C.);

**Keywords:** active constituent, *Pueraria lobata*, potential pharmacodynamic ingredients, type 2 diabetes

## Abstract

**Objectives:** Traditional Chinese Medicine (TCM) is recognized for its complex composition and multiple therapeutic targets. However, current pharmacological research often concentrates on extracts or individual components. The former approach faces numerous challenges, whereas the latter oversimplifies and disregards the synergistic effects among TCM components. This study aims to investigate the scientific validity of focusing on the active constituent in TCM efficacy research, using *Pueraria lobata* (*P. lobata*) as a case study. **Methods:** Through spectrum-effect correlation analysis, network pharmacology, and molecular docking, five active ingredients of *P. lobata* were identified: puerarin, formononetin, tuberosin, 4′,7-dihdroxy-3′-methoxyisoflavone, and Daidzein-4,7-diglucoside. These ingredients were combined to form an active constituent, which was subsequently tested in vitro and in vivo. **Results:** In in vitro, the active constituent exhibited superior effects in enhancing glucose consumption and glycogen synthesis compared to both the *P. lobata* extract and individual components. In vivo experiments demonstrated that medium and high doses of the active constituent were significantly more effective than *P. lobata* extract, with effects comparable to those of metformin in reducing blood sugar levels. **Conclusions:** The active constituent effectively improves T2DM by lowering blood glucose levels, promoting glycogen synthesis, and modulating glycolipid metabolism. Both in vitro and in vivo studies indicate that it outperformed the *P. lobata* extract and individual components. This study establishes the scientific validity and feasibility of utilizing the active constituent as the focus for investigating the efficacy of TCM, thereby offering novel insights and a new research paradigm for future TCM investigations.

## 1. Introduction

Diabetes mellitus (DM) has become increasingly acknowledged as a major health issue in the 21st century, impacting millions of people worldwide. Type 2 diabetes (T2DM) represents more than 90% of all diagnosed diabetes instances, highlighting its significance in conversations about public health. Characterized as a metabolic disorder, type 2 diabetes mellitus is particularly notable for its features, including chronic inflammation, insulin resistance, and the gradual dysfunction of pancreatic β-cells responsible for insulin secretion. As the disease progresses, it generally transitions from an initial phase dominated by insulin resistance—albeit with some residual insulin production—to a later stage marked by significant insulin insufficiency, alongside persistent insulin resistance [1]. Presently, the primary strategies for the prevention of T2DM among patients emphasize education, dietary adjustments, and physical activity. Lifestyle modifications, coupled with pharmacotherapy and patient education, form the cornerstone of treatment for individuals suffering from T2DM. Despite the broad application of various antidiabetic medications in clinical practice, several of these drugs are associated with substantial adverse effects, thereby impeding the effective control of blood glucose levels within the normal range [2,3]. Consequently, the search for antidiabetic agents that are both safer and more effective continues to be a critical focus [4].

Centuries of Traditional Chinese Medicine (TCM) practice have unequivocally demonstrated its therapeutic efficacy [5]. *Pueraria lobata* (Wild.) Ohwi. (*P. lobata*), a Traditional Chinese Medicine (TCM), has been used in clinical practice for thousands of years and is referenced in the Shennong Ben Cao Jing, which is recognized as the earliest authoritative pharmacological text in China. This plant is rich in starch, proteins, and a variety of biologically active compounds, including puerarin, terpenoids, coumarins, and glycosides derived from puerarin [6,7,8]. Contemporary pharmacological research has indicated that extracts from the root and leaves of *P. lobata* exhibit numerous therapeutic effects, particularly in treating cardiovascular disorders, liver injury, osteoporosis, cancer, inflammation, and hyperglycemic abnormalities [9,10]. It has been reported in studies involving hyperglycemia that *P. lobata* can significantly lower blood glucose levels [11].

Modern research indicates that the therapeutic efficacy of Traditional Chinese Medicine (TCM) primarily stems from the overall synergistic effects of its components and their systematic regulation of multiple targets. However, contemporary studies on the pharmacological substances of TCM predominantly concentrate on extracts or isolated single components [12,13]. The study of extracts poses significant difficulties and challenges due to the complexity of their constituent substances. Nonetheless, research that exclusively focuses on a single component oversimplifies the process, disregarding the holistic and synergistic effects of multiple components in TCM, thereby impeding the drug from achieving its intended therapeutic impact. Consequently, we hypothesize that the active constituent (AC) derived from the combination of multiple active ingredients can serve as the focal point for efficacy research in TCM. This approach not only mitigates the challenges associated with studying TCM’s efficacious substances but also accounts for the synergistic effects inherent in its multiple components.

In this study, T2DM was employed as the disease model to investigate the active constituent of *P. lobata* in the intervention of T2DM. Initially, the potential active ingredients in the *P. lobata* extract for the treatment of T2DM were identified through a combination of three methods: spectrum-effect correlation analysis, network pharmacology, and molecular docking. Subsequently, several of the identified potential active ingredients were administered either separately or in combination to assess their therapeutic effects on T2DM. The purpose of this work is to explore the scientific validity and feasibility of utilizing the active constituent as the focal point for investigating TCM efficacy, thereby offering a novel insight and a research paradigm for future TCM investigations.

## 2. Results

### 2.1. Spectrum-Effect Correlation Analysis

UPLC-Q-TOF-MS was utilized to identify the constituents absorbed into the blood of rats in the PUE group. The specific information identified by the UNIFI scientific system is presented in Table 1. The grey correlation analysis was used to correlate the response values of each component with various pharmacological indicators. As shown in Figure 1, a total of 12 components have a correlation degree greater than 0.7, so it is considered that these 12 components may be the potential active components of *P. lobata* for intervention in T2DM.

### 2.2. Network Pharmacology

#### 2.2.1. Gene Collection

A comprehensive collection of disease targets associated with T2DM was achieved, totaling 16,984 targets sourced from databases, including GeneCards, OMIM, and TTD. According to the results of the previous spectral correlation analysis, chemical information of 12 active ingredients was confirmed through the PubChem database. After the removal of duplicates, 270 disease targets affected by PUE were identified. By overlapping the predicted set of 270 potential targets with the 16,984 T2DM disease gene set, we pinpointed the potential targets of PUE relevant to T2DM treatment, and these target data were verified through the UniProt database. Ultimately, data for 167 PUE targets pertinent to the treatment of T2DM were secured, as illustrated in Figure 2A. Additionally, a direct network illustrating the interactions between active ingredients and targets of PUE was developed using Cytoscape 3.7.2, as depicted in Figure 2B.

#### 2.2.2. PPI Network Visualization and Key Target Discovery

A total of 167 genes associated with PUE treatment for T2DM were loaded into STRING, and a PPI network for these genes was generated using Cytoscape 3.7.2, depicted in Figure 2C. The CytoNCA algorithm was employed to assess various network metrics, specifically focusing on degree, betweenness, and closeness. Through this comprehensive evaluation, the analysis identified and selected the top 16 targets that exhibited the most significant relevance within the network. The key targets were RXRA, EGFR, PIK3CA, ESR1, and NR3C1 as shown in Figure 2D. Network topology parameter analysis was performed by the Cytoscape 3.7.2. As shown in Table 2, puerarin, formononetin, daidzein-4,7-diglucoside, 4′,7-dihdroxy-3′-methoxyisoflavone, tuberosin, pulobatones A, and 6″-O-Acetyldaidzin have a degree greater than ten, indicating that these seven compounds may be the potential active ingredients of *P. lobata* in alleviating T2DM.

#### 2.2.3. GO and KEGG Enrichment Analysis

There were 167 genes included in the gene set for the treatment of T2DM that were associated with PUE and were analyzed through a Gene Ontology (GO) functional enrichment analysis. From these results, the top 20 entries with a *p*-value < 0.05 across each category were identified, highlighting significant biological processes. Notably, these processes included responses to drugs, oxidative stress, and lipopolysaccharides, among other closely related biological activities, as illustrated in Figure 3A.

Additionally, the outcomes from the enrichment analysis revealed that the active components of PUE exert their effects on T2DM primarily through specific significant pathways. These pathways include those related to cancer, nitrogen metabolism, receptor activation in chemical carcinogenesis, and serotonergic synapses. To provide a clearer understanding of how these targets interact with various pathways, a KEGG pathway enrichment network diagram was created, as depicted in Figure 3B. This visualization suggests that PUE may contribute to the management and treatment of T2DM through these identified pathways.

### 2.3. Molecular Docking

Through the analysis of the PPI interaction network, key targets were recognized, and the associated chemical components in the drug were further confirmed using molecular docking methods. The necessary files for this analysis were sourced from the Protein Data Bank (PDB) database, ensuring accuracy and relevance. All receptor files underwent treatment to account for organic compounds, water molecules, and hydrogenated charge distributions. For pre-treatment, PyMOL was utilized, while molecular docking was conducted using AutoDockVina to derive the results for molecular verification. The docking results of 12 active ingredients with five core targets are shown in Figure 4A, and part of the visualization results are shown in Figure 4B.

### 2.4. In Vitro Pharmacological Study of P. lobata in Alleviating T2DM

According to the results of network pharmacology and molecular docking, five components including puerarin, formononetin, tuberosin, Daidzein-4,7-diglucoside, and 4′,7-dihdroxy-3′-methoxyisoflavone were further studied as potential active ingredients for treating T2DM with *P. lobata*.

#### 2.4.1. Study on Cytotoxicity

The impact of the drugs on the proliferation of HepG2 cells was assessed by CCK8. The concentration ranges considered safe for each component are outlined in Figure 5.

#### 2.4.2. Study on Hypoglycemic Activity

A 0.06 IU/mL insulin (INS) solution was utilized to induce HepG2 cells for 24 h, thus establishing an insulin resistance HepG2 (IR-HepG2) model. After the successful establishment of the model, the medium that contained varying concentrations of PUE, each component, active constituent, and the positive control metformin (20 μM) was administered separately. Following a 24-h incubation period, the glucose content in supernatant of HepG2 cells was determined by an automatic biochemical analyzer, and the glucose consumption was calculated. The specific results are depicted in Figure 6A–G. Based on this result, the effect concentrations of each component were confirmed as puerarin (20 μM), formononetin (25 μM), tuberosin (25 μM), Daidzein-4,7-diglucoside (70 μM), 4′,7-dihdroxy-3′-methoxyisoflavone (100 μM), PUE (20 μg/mL), and AC (20 μg/mL). Subsequently, the glucose consumption and glycogen levels of IR-HepG2 cells were assessed at the effect concentrations of the five active components, PUE, and AC. As illustrated in Figure 6H,I, PUE has the potential to markedly enhance glucose utilization and stimulate glycogen production in IR-HepG2 cells. The five active components obtained from the preliminary screening and the active constituent also have the same effect, and the active constituent shows a similar effect to the positive control.

### 2.5. In Vivo Pharmacological Study of P. lobata in Alleviating T2DM

#### 2.5.1. Hypoglycemic Effect on T2DM Mice

Administering streptozotocin (STZ) at a dose of 40 mg/kg along with a high-fat and high-sugar diet may lead to elevated fasting blood glucose (FBG) levels. After ten weeks of treatment, PUE notably lowered FBG levels and the dietary consumption of the mice (Figure 7A–C). Notably, the active ingredient exhibited similar effects in a dose-dependent manner.

The results of the oral glucose tolerance test (OGTT) are illustrated in Figure 7D. Throughout the 240-min experimental duration, all groups exhibited peak blood glucose levels at the 40-min mark, reflecting a characteristic pattern of initially rising blood glucose concentrations followed by a subsequent decline. Notably, the blood glucose levels recorded in the Model were significantly elevated when comparing with the Control, which suggests a marked impairment in glucose tolerance among the mice subjected to diabetes. In contrast, the Control demonstrated a complete recovery of blood glucose levels back to baseline after a period of 240 min following the oral glucose administration. Additionally, when evaluating the administration groups, specifically the Met group, PUE group, and AC groups, blood glucose concentrations were significantly reduced than those observed in the Model. After 240 min of administration, these groups also returned to their initial blood glucose levels. This observation aligns with the results derived from calculating the AUC as represented in Figure 7E. The AUC for the OGTT in the Model was significantly higher than that observed in the Control, whereas the administration group displayed a notable reduction when compared to the Model, with a statistically meaningful difference (*p* < 0.01). Among the treatment groups, the high and middle doses of AC exhibited particularly notable effects on the OGTT outcomes in diabetic mice, highlighting their potential as therapeutic interventions for managing glucose intolerance in such models.

#### 2.5.2. Improvement on Insulin Sensitivity in T2DM Mice

The Model exhibited markedly higher INS levels than Control (Figure 8A). Following treatment with metformin and *P. lobata*, the INS level in mice subjected to these interventions, specifically the Met group, PUE group, and AC groups, showed a notable decrease (*p* < 0.01). Among these groups, both the PUE and AC treatments demonstrated a clear therapeutic effect, indicating their potential benefits in managing INS levels. In addition to INS content, the HOMA-IR and HOMA-β were assessed based on the concentrations of blood glucose and INS (Figure 8B,C). The results showed a rise in HOMA-IR and a decline in HOMA-β in the Model compared to the Control. However, post-treatment with *P. lobata* and metformin resulted in a reversal of these indices, suggesting that *P. lobata* is capable of addressing the damage inflicted on pancreatic islet β cells due to diabetes, effectively restoring their functionality. Notably, the intermediate and high doses of the AC group yielded superior outcomes in this regard.

The insulin tolerance test (ITT) was performed three days before the experiment’s conclusion, as depicted in Figure 8D,E. The results indicated a notable pattern in the blood glucose levels of all groups of mice following the injection of insulin. During the 120 min of the experiment, the Control mice exhibited notably reduced blood glucose levels in comparison to the Model. After 120 min post-insulin injection, the blood glucose levels of the Control returned to baseline. For the diabetic mice that received treatments with *P. lobata* and metformin, the decreases in blood glucose levels were significantly greater than those observed in untreated diabetic mice. This observation underscores the potential therapeutic effects of both *P. lobata* and metformin on glucose metabolism in diabetic subjects. A quantitative evaluation through the AUC for the ITT further supported these findings. The results showed that the AUC_ITT_ for the Model was significantly elevated when compared to the Control (*p* < 0.01), affirming the poorer insulin response in untreated diabetic mice. In contrast, the AUC_ITT_ measurements for the Met group, PUE group, and AC groups were markedly lower than that of the Model (*p* < 0.01), and AC_M and AC_H showed a similar effect to Met. This implies that diabetic mice treated with *P. lobata* and active constituent exhibited increased insulin sensitivity. Together, these results demonstrate that *P. lobata* and active constituent may enhance insulin sensitivity in diabetic mice and positively aid in the regeneration of damaged islet β cells.

#### 2.5.3. Effects of *P. lobata* on Serum Biochemical Indices of T2DM

To investigate the impact of diabetes on the serum biochemical indices of *P. lobata*, we conducted a comprehensive analysis of several key parameters. The levels of total cholesterol (T-CHO), triglycerides (TG), low-density lipoprotein cholesterol (LDL-C), high-density lipoprotein cholesterol (HDL-C), glycated hemoglobin (Ghb), and ketone bodies (KB) in the serum samples collected from mice were measured. As shown in Figure 9, the levels of GHb, LDL-C, T-CHO, and TG were significantly increased (*p* < 0.01) in the Model compared with Control, while the levels of KB and HDL-C were significantly reduced (*p* < 0.01). After administration of PUE and AC, the abnormal changes of biochemical indicators were all reversed. The above results suggested that PUE and AC, similar to metformin, can also improve the glycolipid metabolism disorder of T2DM mice. Among them, the AC showed a better effect in some parameters but not all (three of six were significantly different for AC_L).

#### 2.5.4. Pathological Changes of Liver and Pancreas Tissue in T2DM Mice

The results of histological examination demonstrated that PUE and the active constituent had an obvious amelioration on liver and pancreas morphological change (Figure 10). Herein, we could observe obvious pathological changes such as fat vacuoles, inflammatory cells, hepatocyte necrosis, and severe morphological damage in the liver, and decreased number and volume of pancreatic islets, turbidity, swelling, or vacuolar degeneration of pancreatic islet cells, and inflammatory cell infiltration in pancreas of STZ-induced T2DM mice. The treatment of PUE and AC could significantly attenuate these morphological changes. In addition, PAS staining was used to evaluate the distribution of glycogen in the liver. When the body is in a state of insulin resistance, the synthesis of glycogen in the body is impaired, and its content decreases, leading to an increase in blood glucose content. After PAS staining, the depth of purple represents the amount of glycogen in the body. As illustrated in Figure 10B, the mice of the Model exhibited significant decomposition of liver glycogen when compared to the Control. However, after administration for 10 weeks, the liver glycogen accumulation of mice in the PUE and AC groups increased significantly dependent on concentration, indicating that PUE and the active constituent had a strong improvement effect on the liver glycogen decomposition caused by type 2 diabetes.

## 3. Discussion

Type 2 diabetes mellitus (T2DM) is recognized as a chronic metabolic disorder primarily characterized by elevated blood glucose levels. This condition results from the body’s ineffective utilization of insulin, leading not only to hyperglycemia but also to significant disruptions in lipid metabolism [14]. Over time, these metabolic abnormalities can lead to various complications, rendering T2DM a major public health concern that necessitates careful management and intervention [15]. Therefore, it is crucial to effectively manage both hyperglycemia and hyperlipidemia for the treatment and prevention of T2DM. *Pueraria lobata* (Wild.) Ohwi (*P. lobata*) has been utilized as a Traditional Chinese Medicine (TCM) in clinical practices for millennia. In this study, the active constituent of *P. lobata* in alleviating T2DM was screened, and the efficacy of the active constituent was verified both in vivo and in vitro. This research offers a novel insight and research paradigm for exploring the active ingredients in TCM.

The spectrum-effect correlation entails a correlation analysis between the chemical information of TCM and the efficacy results obtained via statistical methods [16]. In this study, grey relational analysis was used to correlate the blood entering components of *P. lobata* extract with its improvement effect on the pharmacodynamic indicators of T2DM rats. With correlation degree greater than 0.7 as the evaluation criterion [17], 12 active ingredients were obtained, including puerarin, pulobatones B, formononetin, tuberosin, 3′-Hydroxy-6″-*O*-xylosylpuerarin, Daidzein-4,7-diglucoside, 4′,6″-diacetyl puerarin, Corchoionoside C, pulobatones A, kudzusaponin A1, 6″-*O*-Acetyldaidzin, and 4′,7-dihdroxy-3′-methoxyisoflavone which were closely related to T2DM. These 12 ingredients were further analyzed as potential active ingredients for *P. lobata* intervention in T2DM.

In network pharmacology, a comprehensive analysis revealed a total of 270 direct target genes associated with 12 active components found in *P. lobata*. This finding underscores the significant pharmacological potential of *P. lobata* in the management of T2DM. The involvement of multiple targets further suggests that *P. lobata* may exert its therapeutic effects through a complex interaction with various biological pathways, highlighting its potential as a multifaceted treatment option for this metabolic disorder. According to the results of network pharmacology, the degree values of seven components, including puerarin, formononetin, daidzein-4,7-diglucoside, 4′,7-dihdroxy-3′-methoxyisoflavone, tuberosin, pulobatones A, and 6″-*O*-Acetyldaidzin, are all greater than ten, indicating a higher correlation with the core target. Therefore, these seven components can be considered as the active components of *P. lobata* in the intervention of T2DM.

Additionally, five key targets were identified based on the PPI interaction network: RXRA, EGFR, PIK3CA, ESR1, and NR3C1. To further confirm the effect of the active ingredients in the intervention of T2DM, molecular docking was performed with 12 potential active ingredients against these five key targets. Research indicates that a binding energy of less than 0 kJ/mol between the ligand molecule and the receptor protein suggests a natural association. Furthermore, a binding energy below −5 kJ/mol indicates a more favorable binding interaction, with lower binding energies correlating to improved docking outcomes [18]. Among the 60 docking results obtained, all compounds and proteins demonstrated the ability to bind spontaneously, with most exhibiting binding energies less than −5 kJ/mol and some below −10 kJ/mol, suggesting enhanced docking performance. In the molecular docking analysis, six components—Daidzein-4,7-diglucoside, puerarin, formononetin, 4′,7-dihydroxy-3′-methoxyisoflavone, pulobatones B, and tuberosin—showed a stronger affinity for the five core targets, indicating that these components may represent the active ingredients of *P. lobata* in alleviating T2DM.

Combined with the above three methods, five ingredients, including puerarin, formononetin, Daidzein-4,7-diglucoside, 4′,7-dihdroxy-3′-methoxyisoflavone, and tuberosin were finally confirmed as the active ingredients of *P. lobata* intervention in T2DM for further study. In order to investigate the pharmacodynamic effect of the active constituent, five active components were mixed in equal proportion as the active constituent of *P. lobata* and the pharmacodynamic study was carried out in vivo and in vitro.

Insulin resistance (IR) is a critical factor in the development of hyperglycemia and T2DM, as well as various metabolic irregularities, including elevated blood sugar levels, dyslipidemia, and increased release of inflammatory agents [19]. In the in vitro pharmacodynamic study, an IR-HepG2 cell model was established, and the effective concentration of *P. lobata* extract, along with five active components and the active constituent, was screened based on glucose consumption. Following the administration of the drug, glucose uptake and utilization exhibited a significant increase. This finding suggests that the extract of *P. lobata*, along with its five active components and the active constituent, effectively enhances glucose utilization in IR-HepG2 cells. Earlier research has shown that impaired glycogen synthesis and a failure to regulate glucose production in hepatocytes are primarily outcomes of insulin resistance [20]. Our findings demonstrated that IR-HepG2 cells exhibited a significant decrease in intracellular glycogen levels; however, treatment with *P. lobata* extract, its five active components, and the active constituent resulted in an increase in glycogen content in these cells. This indicates that *P. lobata* facilitates glycogen synthesis in IR-HepG2 cells. Importantly, the active constituent exhibited stronger activity than *P. lobata* extract and the five active components when used individually, showing effects comparable to those of the positive control, metformin.

In the in vivo pharmacodynamic study, an STZ-induced T2DM mouse model was established. Over a period of 10 weeks, we observed alterations in dietary consumption and blood glucose levels in the mice. Typical signs of diabetes, such as increased food and water intake, were noted [21,22]. Following the administration of the *P. lobata* extract and its active constituent, the food and water consumption in the treated mice was reduced compared to that of the diabetic group, suggesting an alleviation of these symptoms. Additionally, the results indicated that the *P. lobata* extract and its active constituent could significantly reduce FBG levels in a dose-dependent manner.

IR is a detrimental physiological condition characterized by the insensitivity of the body’s cells to insulin, which impairs the hormone’s functionality. Enhancing the sensitivity to insulin can improve its effectiveness, thereby facilitating glucose metabolism and utilization. Research indicates that insulin level in T2DM mice is significantly elevated compared to normal mice. This prolonged hyperinsulinemic state may lead to detrimental effects, exacerbating symptoms of IR and impairing insulin tolerance. Assessments using the HOMA-IR and the HOMA-β reveal that T2DM mice exhibit severe insulin resistance accompanied by damage to islet β cells [23]. After treatment with *P. lobata* extract and its active constituent, a reduction in the HOMA-IR and an increase in the HOMA-β were observed, indicating alleviation of insulin resistance symptoms. Moreover, results from the ITT demonstrated that both *P. lobata* extract and the active constituent effectively enhance insulin sensitivity in T2DM mice. These findings suggest that *P. lobata* extract and the active constituent are capable of lowering blood glucose levels by modulating insulin resistance and tolerance. Notably, administration of middle and high doses of the active constituent yielded superior effects compared to the *P. lobata* extract, and the active constituent exhibited efficacy comparable to that of the positive control, metformin.

Dyslipidemia is a major manifestation of T2DM [24]. Glycolipid metabolisms are intricately interconnected within the human body. When glucose metabolism is dysfunctional, lipid metabolism is also compromised. Consequently, T2DM patients frequently exhibit dyslipidemia, which can lead to hyperlipidemia [25]. The lipid profiles in the blood of mice suffering from T2DM were evaluated, with particular emphasis on levels of T-CHO, TG, HDL-C, and LDL-C. Findings revealed that following treatment with *P. lobata* extract and its active constituent, there was a marked reduction in the blood lipid levels among T2DM mice, alongside an increase in HDL-C, supporting this observation. Although the ability of the AC groups to regulate blood lipids in T2DM mice was slightly inferior to that of metformin, the effects of AC_M and AC_H were still superior to those observed with the *P. lobata* extract.

The liver plays an important role in the metabolism of both lipids and glucose. Damage to the liver in mice can result in dyslipidemia, which may manifest as IR in the liver and disrupted glycogen synthesis [26]. Pathological changes in the liver and pancreas of the model were observed through hematoxylin and eosin (H&E) staining, with these lesions alleviated following treatment. The findings suggest that the *P. lobata* extract, along with its active constituent, has the potential to enhance glycolipid metabolism by facilitating the repair of liver cells. Furthermore, the liver functions as a central hub for glycogen synthesis and storage, playing a vital role in maintaining overall metabolic balance. To assess the impact of *P. lobata* extract on liver glycogen levels, Periodic Acid-Schiff (PAS) staining was utilized. The results of this evaluation demonstrated a significant increase in glycogen accumulation within the livers of mice with T2DM that were treated with *P. lobata* extract and its active constituent. This finding underscores the efficacy of the active constituent in promoting liver glycogen synthesis, which, in turn, aids in the regulation of blood glucose levels. Such outcomes highlight the therapeutic potential of the *P. lobata* extract and its active constituent in managing metabolic disorders, particularly in controlling hyperglycemia.

In the aforementioned pharmacodynamic studies, the active constituent derived from the equal mixture of the five active ingredients of *P. lobata* exhibited a robust effect in alleviating T2DM. The active constituent outperformed both the *P. lobata* extract and the individual use of the five active ingredients, exhibiting an effect comparable to that of metformin in reducing blood glucose levels. These findings substantiate the scientific validity and feasibility of employing active constituent as the focal point for investigating the efficacy of TCM. Concomitantly, future research should focus on optimizing the content ratios of each component in the active constituent to enhance its therapeutic efficacy.

## 4. Materials and Methods

### 4.1. Materials

*Pueraria lobata* (Willd.) Ohwi was obtained from the Bozhou Baopu Pharmaceutical Co. Ltd. (Bozhou, China). All samples were authenticated by Long Wang (School of Traditional Chinese Pharmacy, China Pharmaceutical University, China). Puerarin (B20446, purity ≥ 98%), daidzein-4,7-diglucoside (B52554, purity ≥ 98%), 4′,7-dihydroxy-3′-methoxyisoflavone (B31005, purity ≥ 98%), formononetin (B20836, purity ≥ 98%) and tuberosin (B52477, purity ≥ 98%) were purchased from Shanghai Yuanye Biotechnology Co., Ltd. (Shanghai, China). Streptozocin (STZ, C1616118, purity ≥ 98%) and Metformin hydrochloride (Met, I1904088, purity ≥ 98%) were obtained from Shanghai Aladdin Bio-Chem Technology Co., LTD. (Shanghai, China). Sodium citrate and citric acid were purchased from Sinopharm Chemical Reagent Co., Ltd. (Shanghai, China). ACCU-CHEK Active Blood Glucose Test Strips were ordered from Roche Diagnostics Co., Ltd. (Shanghai, China). High-fat and high-sugar feed was purchased from Jiangsu Synergy Medicine Bioengineering Co., Ltd. (Nanjing, China). TG, T-CHO, LDL-C, and HDL-C kits were all obtained from Nanjing Jiancheng Bioengineering Institute (Nanjing, China). INS, GHb, KB, and IRS-1 ELISA kits were obtained from Jiangsu Meimian Industrial Co., Ltd. (Yancheng, China).

### 4.2. Preparation of the P. lobata Extract

The roots of *P. lobata* (1 kg) were mashed and combined with an 80% ethanol aqueous solution at a ratio of 1:10 (g/mL). This mixture underwent reflux extraction at a temperature of 95 ± 2 °C for 2 h, followed by filtration to gather the filtrate. This process was repeated two additional times. Subsequently, the filtrates were subjected to vacuum-rotary evaporation and freeze-drying, resulting in the acquisition of 221.26 g of the extract from *P. lobata* (PUE).

### 4.3. Spectrum-Effect Correlation Analysis

#### 4.3.1. Animal Models

Twenty-four SD male rats weighing 180–220 g were purchased from Oinglongshan Animal Breeding Farm (license number: SCXK (Zhe) 2021–0192, Nanjing, China) and housed in the barrier environment of the New Animal Center of China Pharmaceutical University at 23 ± 2 °C; the humidity was maintained at 45 ± 10%, under a 12 h light–dark cycle with access to water and rodent chow ad libitum. Approval for all experimental procedures was obtained from the Laboratory Animal Ethics and Management Committee at China Pharmaceutical University, in full compliance with relevant international, national, and institutional standards for animal care and use. The experiment was conducted meticulously, adhering to the International Guidelines for Animal Welfare, and the assigned identification number was 2021-12-026.

SD rats (n = 24) were randomly divided into 4 groups: control group (Control), model group (Model), positive control group (Met, 200 mg/kg/d), and *P. lobata* extract group (PUE, 700 mg/kg/d) (n = 6). The rats in Control were maintained on normal feed and the rats in other groups were maintained on high-fat and high-sugar diets to accelerate the onset of diabetes. The rats were given food for 4 weeks, followed by a fasting period of 12 h. The rats in both the model and PUE groups received intraperitoneal injections of 40 mg/kg STZ (prepared in a sodium citrate buffer solution of 0.1 mol/L at pH 4.4 in an ice bath) over the course of 3 days, whereas rats in the Control group were administered an equivalent volume of the sodium citrate buffer solution. Diabetic rats were classified as those exhibiting hyperglycemia, defined by FBG ≥ 11.1 mmol/L [27]. Then, the rats in Control and Model were administrated water, while the rats in Met and PUE were given metformin hydrochloride and *P. lobata* extract, respectively. Rats in each group were orally administered continuously for 4 weeks.

#### 4.3.2. Measurement of Biochemical Indicators

The rats were fasted for 12 h after the end of administration, and blood was collected from the abdominal aorta after anesthesia. Centrifugation of the blood was performed at 3000 r/min for a duration of 10 min, after which the serum was isolated for subsequent experiments. The serum was divided into two parts, one for detecting the blood components of PUE and the other for detecting the biochemical indicators. INS, TG, LDL-C, HDL-C, T-CHO, GHb, and KB values were measured according to the protocols of manufacturer.

#### 4.3.3. UPLC-Q/TOF-MS/MS Analysis

Metabolomics analysis was accomplished using UPLC-Q-TOF/MS with electrospray ionization (ESI) in negative ion mode, collected using Model, Control, and PUE, respectively. Leucine enkephalin and sodium formate were used as the lock mass. The parameters for optimization included a sample cone voltage of 40 V; a capillary voltage set to 2.5 kV; a source temperature maintained at 100 °C; an extraction cone voltage of 40 V; and desolvation gas settings of 450 °C for temperature and 800 L/h for flow rate, along with a cone gas flow of 50 L/h. The low collision energy was established at 6 V, while the ramp collision energy varied between 30 and 70 V during MSE scan mode detection. Data acquisition spanned an *m*/*z* range from 50 to 1200 with a scanning interval of 0.2 s.

Given the variations in retention time and elution sequence that can occur during the UPLC-Q-TOF/MS analysis process, it is crucial to maintain consistent monitoring of the system’s performance. In the current study, we prepared a quality control (QC) sample by mixing 50 µL of serum samples, which served as a benchmark for method validation. Initially, the QC sample was examined three times at the beginning of the entire sample series, and thereafter, it was injected following every four samples during the analytical run. This approach was implemented to ensure the reliability and consistency of the system during the analysis.

#### 4.3.4. Grey Relational Analysis

As the first step in spectrum-effect correlation, the UPLC-Q-TOF/MS conditions were the same as described previously. INS, TG, LDL-C, HDL-C, T-CHO, GHb, and KB values were measured according to manufacturer’s protocols. Furthermore, grey relational analysis (GRA) was conducted by the Data Processing System (DPS V15.10) to evaluate the link between the compounds and their pharmacodynamic effects. Prior to executing the GRA, the peak response intensities were standardized. The standardized data served as the subsequences, while the levels of serum indicators were designated as the main subsequence [28]. The relative correlation between ingredients and drug efficacy is the average correlation between ingredients and various serum indicators.

### 4.4. Network Pharmacology

#### 4.4.1. Gene Collection of T2DM

Genes associated with T2DM were retrieved from several databases: Online Mendelian Inheritance in Man (OMIM) [29], GeneCards [30], PharmGkb [31], and DrugBank [32]. Within these resources, the relevant T2DM genes were identified using the input term “Type 2 Diabetes”.

#### 4.4.2. Prediction and Identification of *P. lobata* Components and Potential Targets

Information concerning the chemical name, molecular formula, PubChem Compound Identifier (CID), and Canonical SMILES for each individual constituent was collected from the PubChem database, accessible at https://pubchem.ncbi.nlm.nih.gov/ (accessed on 11 January 2022). The TCMSP database was employed to enhance the prediction of targets associated with the active components, ultimately yielding the related targets for the effective components of *P. lobata*. A comparative analysis was performed between the anticipated potential target collection and the gene set associated with T2DM to identify the possible target set of *P. lobata* for T2DM treatment, with the target details confirmed through the UniProt database (https://www.uniprot.org/, accessed on 13 January 2022). In conclusion, the component-target data relevant to *P. lobata* for T2DM treatment were compiled. Employing Cytoscape (Version: 3.7.2), an active ingredient-target network for *P. lobata* in T2DM treatment was illustrated. For further analysis, this network was designated as the direct target network concerning the active constituent of *P. lobata*.

#### 4.4.3. Protein–Protein Interaction (PPI) Network Analysis

To enhance the investigation and pinpoint the primary targets within the treatment network of T2DM using *P. lobata*, this study evaluates significant targets in the PPI network through the lens of network modules. The CytoNCA tools that are integrated into the Cytoscape software (3.7.2) were utilized to evaluate the various network topology metrics associated with the targets. These metrics included degree, betweenness, and closeness, which are crucial for understanding the structure and dynamics of the network [33].

#### 4.4.4. GO and KEGG Enrichment Analysis

The target set of *P. lobata* for T2DM treatment was analyzed for GO functional annotation and KEGG pathway enrichment using ClusterProfiler (Version: 3.14.3). The procedure for choosing directories for gene annotation and signaling pathways, intended for functional annotation using GO and for enrichment analysis of pathways in the KEGG, was conducted based on defined statistical criteria. These criteria included a *p*-value of less than 0.05 and a q-value also below 0.05.

### 4.5. Molecular Docking

3′-hydroxy-6″-*O*-xylosylpuerarin, pulobatones A, tuberosin, pulobatones B, puerarin, formononetin, 4′,7-dihdroxy-3′-methoxyisoflavone, Daidzein-4,7-diglucoside, 4′,6″-diacetyl puerarin, 6″-*O*-Acetyldaidzin, kudzusaponin A1, and corchoionoside C were selected as the potential active components of *P. lobata* in type 2 diabetes. The core target proteins of *P. lobata* in type 2 diabetes were screened by network pharmacology. Search PubChem database (https://PubChem.ncbi.nlm.nih.gov/, accessed on 11 January 2022) for CAS numbers of small molecules, download the SDF format of the corresponding compounds, and use Open Babel 3.1.1 to convert to MOL2 format for docking preparation. From the PDB database (https://www.rcsb.org/, accessed on 17 January 2022), the 3D structure of the core protein was obtained by screening for species, conformational resolution, sequence integrity, and pH, and stored as a PDB file for docking. Autodocktools 1.5.7 and Pymol 2.5 were used to dehydrate, hydrogenate and select as the receptor, hydrogenate and select as the ligand, and twist the bonds and centers of the active compounds and the core target proteins. After running AutoGrid and AutoDock, we obtained the docking results of the compounds and proteins, made a heat map, and visualized them using Pymol 2.5 software.

### 4.6. Pharmacological Study of Active Constituent of P. lobata in the Treatment of T2DM

#### 4.6.1. Cell Culture

HepG2 cells, originating from human hepatocellular carcinoma of the liver, were obtained from the Cell Resource Center based in Beijing, China. These cells were cultured in a high glucose DMEM (Gibco BRL, Grand Island, NY, USA) supplemented with 10% fetal bovine serum (Gibco, USA) and 1% antibiotics (100 μg/mL penicillin and 100 μg/mL streptomycin, New Cell & Molecular Biotech Co., Ltd., Guizhou, China). The incubation was maintained at 37 °C with 5% CO_2_, and the culture medium was replaced three times weekly.

#### 4.6.2. Cell Viability

Cell survival was evaluated in HepG2 cells with the Cell Counting Kit (CCK8, Bio-Channel, Nanjing, China). Initially, cells with a density of 2 × 10^4^ cells per well were sown in a 96-well microplate and incubated for 24 h. After this incubation period, the cells underwent treatment with various concentrations of PUE (5, 10, 15, 20, 25, 30, 35, and 40 μg/mL), five active components (2, 4, 8, 16, 32, 64, 128, and 256 μM), and the active constituent (AC) at concentrations of 5, 10, 15, 20, 25, 30, 35, and 40 μg/mL for an additional 24 h in a 5% CO2 atmosphere, maintained at 37 °C. The control group was held under the same experimental conditions. Each treatment group consisted of six replicate wells. After processing, 10 μL of CCK8 detection solution was added to each borehole. Then the sample was incubated at 37 °C for two hours and absorption was measured at 450 nm with a multifunctional microplate reader (Thermo Fisher Scientific, Waltham, MA, USA).

#### 4.6.3. Glucose Consumption Assay

The method of glucose oxidase-peroxidase was employed to determine glucose consumption. Cells were transferred to a 96-well plate at a density of 2 × 10^4^ cells per well, with 6 wells left as blanks. After a period of 24 h of treatment, the culture medium was collected for analysis. Subsequently, the glucose level present in the supernatant was measured to determine its concentration. To accurately calculate the consumption of glucose during the experiment, the concentration of glucose found in the control well was subtracted from the concentration measured in the experimental well.

#### 4.6.4. Glycogen Assay

HepG2 cells in the logarithmic growth phase were subjected to treatment with 0.25% trypsin-EDTA (Gibco, USA) to generate a suspension of single cells. Following this, the cells were centrifuged at 1000 rpm for 10 min, and the supernatants were subsequently discarded. Subsequently, the cells were subjected to sonication in an ice water bath. The alkaline solution was added to the sample without centrifugation and soaked in boiling water for 20 min to obtain the sugar detection solution. Glycogen value was measured according to manufacturer’s protocols.

#### 4.6.5. Animal Models

Eight-week-old male C57BL/6 mice, each weighing 20 ± 2 g, were obtained from the Nanjing Branch of Beijing Vital River Laboratory Animal Technology Co., Ltd. (license number: SCXK (Zhe) 2022-0002, Beijing, China). C57BL/6 mice were housed in Experimental Animal Center of China Pharmaceutical University at 23 ± 2 °C and relative humidity was maintained at 45 ± 10%. The mice were exposed to a 12-h cycle of light and darkness and had unrestricted access to both water and rodent food. All procedures carried out in this study were in compliance with the Laboratory Animal Ethics and Management Committee’s requirements, reflecting a commitment to all relevant international, national, and institutional guidelines governing animal welfare, having received approval from the Animal Ethics Committee of China Pharmaceutical University. The experiment was conducted diligently according to the International Guidelines for Animal Welfare, and the approved identification number for this study is 2022-03-024.

A total of 42 C57BL/6 mice were allocated at random into seven distinct groups: the control group (Control), the model group (Model), the positive control group (Met, 100 mg/kg/day), the group receiving *P. lobata* extract (PUE, 1 g/kg/day), and groups for the active constituent (the equal mixture of the 5 active ingredients of *P. lobata*, AC_L, 50 mg/kg/day; AC_M, 100 mg/kg/day; AC_H, 200 mg/kg/day), with each group consisting of six mice (n = 6). The mice in the Control group received a standard diet, while those in the other groups were fed a diet rich in fats and sugars to hasten the development of diabetes. The mice were fed for 4 weeks and then fasted for 12 h. Mice assigned to the Model and treatment groups received intraperitoneal (i.p.) injections of 40 mg/kg STZ (prepared in a 0.1 mol/L sodium citrate buffer solution at pH 4.4 in an ice bath) over a period of 3 days. In contrast, mice in the Control group received an equivalent volume of sodium citrate buffer by itself. Mice with hyperglycemia (FBG ≥ 11.1 mmol/L) could be defined as diabetic mice. Following a 10-week period of gavage, each group of mice underwent an 8-h fasting period before blood samples were retrieved from the orbital vein. Afterward, the mice were euthanized via the cervical dislocation technique, and their livers and pancreases were harvested for additional research.

#### 4.6.6. Oral Glucose Tolerance Test (OGTT) and Insulin Tolerance Test (ITT)

The food and water taken by the mice was recorded every day and FBG values of mice were assessed by ACCU-CHEK Active Blood Glucose Test Strips every week. At the end of the 10th week, the OGTT was carried out. Following the treatment, the mice underwent a fasting period of 12 h overnight in preparation for the OGTT experiments. On the subsequent morning, the mice were given an oral dose of glucose at a concentration of 2 mg/kg. Blood samples were then collected from the tail veins of the mice at specific time intervals: 0, 40, 80, 120, 160, 200, and 240 min. These samples were analyzed to measure the blood glucose levels, which allowed for the creation of a glucose response curve for the OGTT. Additionally, the AUC was calculated to provide a quantifiable measure of the mice’s glucose tolerance response over the designated time frame.

To evaluate IR among different groups of mice, the ITT was administered. The test was performed following a 12-h fasting period. Blood samples were taken from the tail of mice and their blood glucose levels were measured with an ACCU-CHEK active glucose meter (F. Hoffman La Roche GmbH, Basel, Switzerland) to determine the basic blood glucose. Each mouse was administered human insulin at a dosage of 1 IU/kg via intraperitoneal injection. Blood samples were taken at time points of 0, 30, 60, 90, and 120 min to assess blood glucose levels. The blood glucose values were used to construct an ITT curve, and the AUC was measured.

#### 4.6.7. Analysis of Serum Biochemical Indicators

Mouse serum T-CHO, TG, LDL-C, HDL-C, and GHB concentrations were evaluated with commercial kits from the Nanjing Institute of Bioengineering in China. Each work step was carried out strictly according to the manufacturer’s instructions.

#### 4.6.8. Determination of Insulin Sensitivity

Blood samples were collected from the tail vein of mice and their blood glucose levels were evaluated through a blood glucose meter. Serum insulin (INS) in mice was quantified using an ELISA kit provided by Jiangsu Meimian Industrial Co., Ltd. (Yancheng, China), following the manufacturer’s instructions diligently. The formula for evaluating insulin resistance through the homeostasis model (HOMA-IR) is as follows: HOMA-IR = [fasting blood glucose (mmol/L) × fasting plasma insulin (mIU/L)]/22.5. This equation integrates measurements of plasma insulin levels and fasting blood glucose to yield an estimation of insulin resistance in subjects. Conversely, the calculation for the homeostasis model assessment of insulin sensitivity (HOMA-β) is described by the following formula: HOMA-β = 20 × fasting plasma insulin (mIU/L)/fasting blood glucose (mmol/L) − 3.5. This calculation serves to evaluate insulin sensitivity by relating fasting insulin levels to fasting blood glucose levels, enabling a better understanding of an individual’s metabolic state.

#### 4.6.9. Histological Examination

After gathering samples, small pieces of mouse liver tissue and the pancreas were preserved in 4% paraformaldehyde for 24 h, followed by embedding and sectioning [34]. Subsequently, histological changes of the liver were observed by H&E staining and Periodic Acid-Schiff staining (PAS) under a 400× microscope, and the changes of the pancreas were observed by H&E staining under a 100× microscope.

### 4.7. Statistical Analysis

Statistical analysis was conducted utilizing SPSS software version 22.0 (IBM Corp., Armonk, New York, NY, USA). In accordance with the assessments of normality and homogeneity of variance, either one-way ANOVA or the rank sum test was employed for the analysis of the data, with a *p*-value of less than 0.05 deemed statistically significant.

## Figures and Tables

**Figure 1 pharmaceuticals-17-01675-f001:**
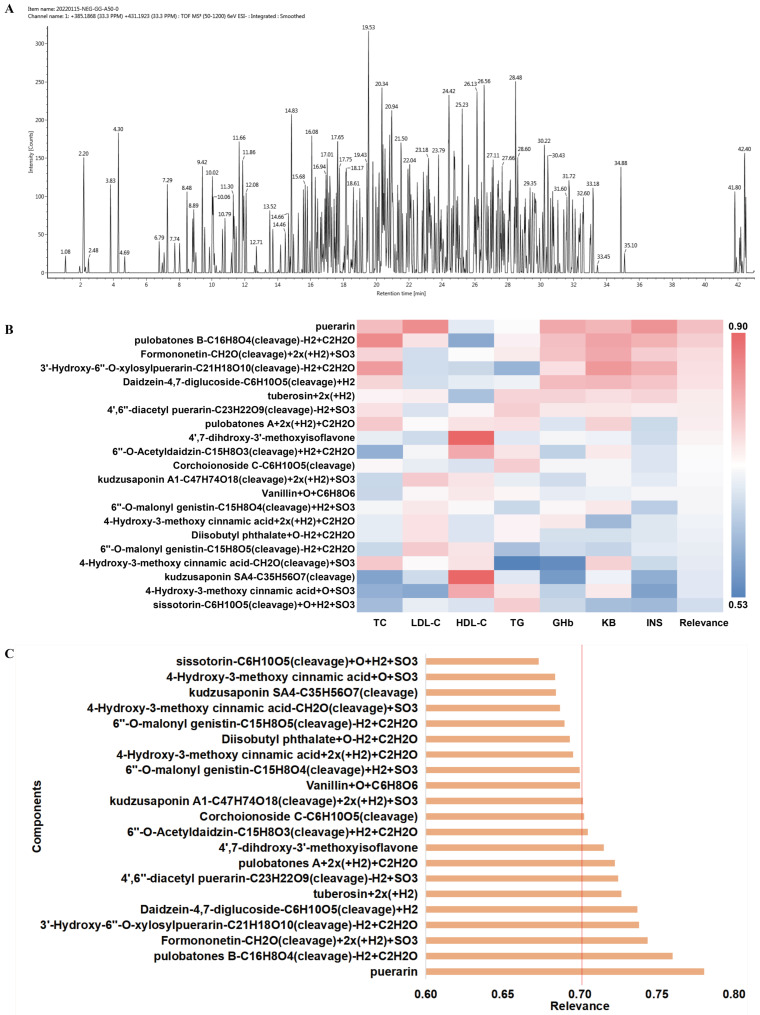
Results of spectrum-effect correlation. (**A**) Total ions chromatogram of the blood components of *P. lobata* extract in rats; (**B**) Heat map of the correlation between the blood components of *P. lobata* and the serum indicators; (**C**) The correlation between the components of *P. lobata* and comprehensive evaluation indicator, the red line indicates a correlation of 0.70.

**Figure 2 pharmaceuticals-17-01675-f002:**
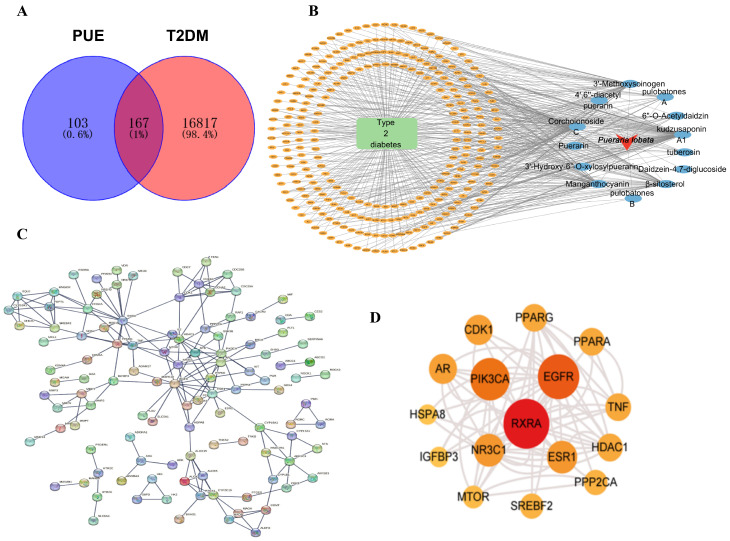
(**A**) The shared targets of *P. lobata* in the TCMSP and TCMID databases, along with differential genes associated with T2DM from the GeneCards, OMIM, and TTD datasets; (**B**) The compound-target network illustrating 12 candidate active ingredients and their 167 potential targets for *P. lobata* in T2DM; (**C**) The PPI network of genes involved in the treatment of T2DM with *P. lobata*; (**D**) Outcomes representing the top 16 significant targets within the PPI network as determined by CytoNCA.

**Figure 3 pharmaceuticals-17-01675-f003:**
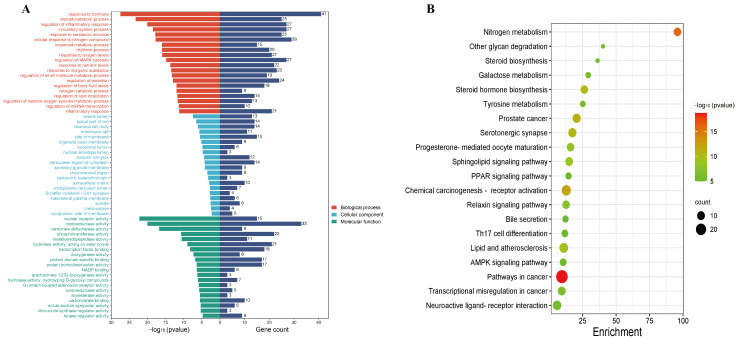
(**A**) GO analysis for 167 protein targets in the treatment of T2DM with *P. lobata*; (**B**) KEGG pathway analysis involving 167 protein targets related to *P. lobata* therapy for T2DM.

**Figure 4 pharmaceuticals-17-01675-f004:**
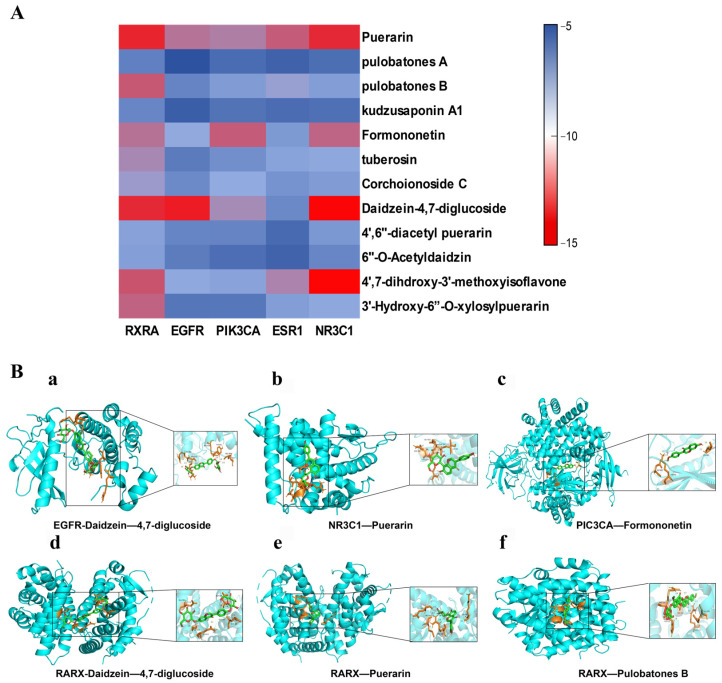
Results of the molecular docking. (**A**) Heat map of the docking binding energy between 12 active compounds and five core targets; (**B**) Pattern diagram of molecular docking.

**Figure 5 pharmaceuticals-17-01675-f005:**
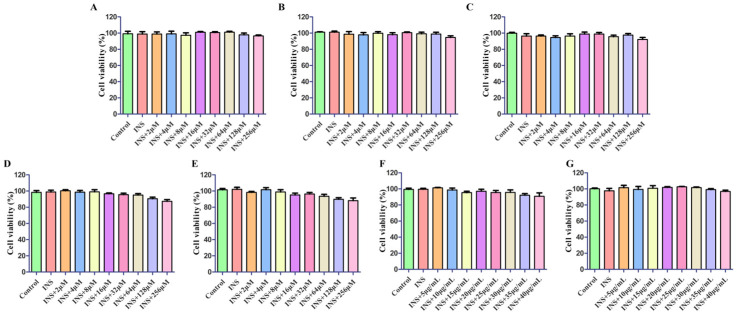
Cell viability in IR-HepG2 cells induced by INS. Puerarin (**A**), formononetin (**B**), tuberosin (**C**), Daidzein-4,7-diglucoside (**D**), 4′,7-dihdroxy-3′-methoxyisoflavone (**E**), *P. lobata* extract (**F**), and active constituent (**G**).

**Figure 6 pharmaceuticals-17-01675-f006:**
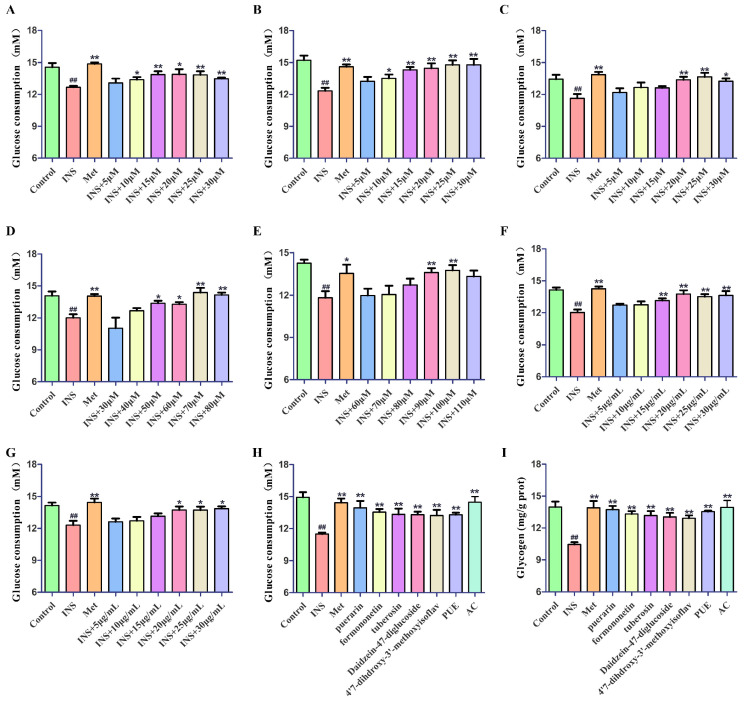
Beneficial effect curve of puerarin (**A**), formononetin (**B**), tuberosin (**C**), Daidzein-4,7-diglucoside (**D**), 4′,7-dihdroxy-3′-methoxyisoflavone (**E**), *P. lobata* extract (**F**), and active constituent (**G**); effect of *P. lobata* on glucose consumption (**H**) and glycogen levels (**I**) in IR-HepG2 cells. In comparison to the control group, ^##^
*p* < 0.01; when contrasted with the model group, * *p* < 0.05, ** *p* < 0.01.

**Figure 7 pharmaceuticals-17-01675-f007:**
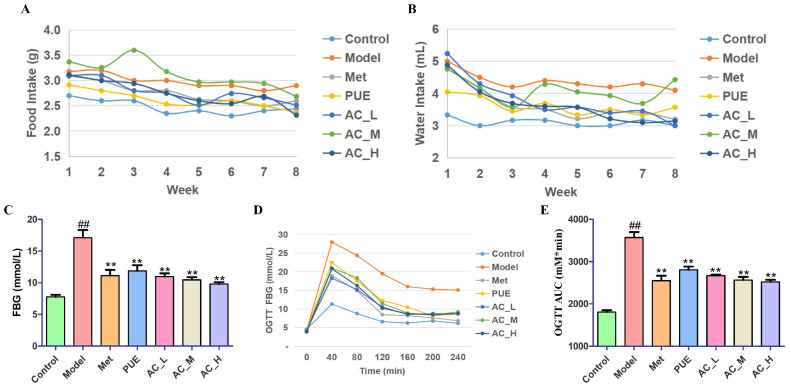
Hypoglycemic effect of *P. lobata* extract and active constituent on STZ-induced T2DM mice. Food intake (**A**), water intake (**B**), FBG levels (**C**), and OGTT (**D**,**E**) of the mice treated with *P. lobata* extract and active constituent for 10 weeks. In comparison to the control group, ^##^
*p* < 0.01; when contrasted with the model group, ** *p* < 0.01.

**Figure 8 pharmaceuticals-17-01675-f008:**
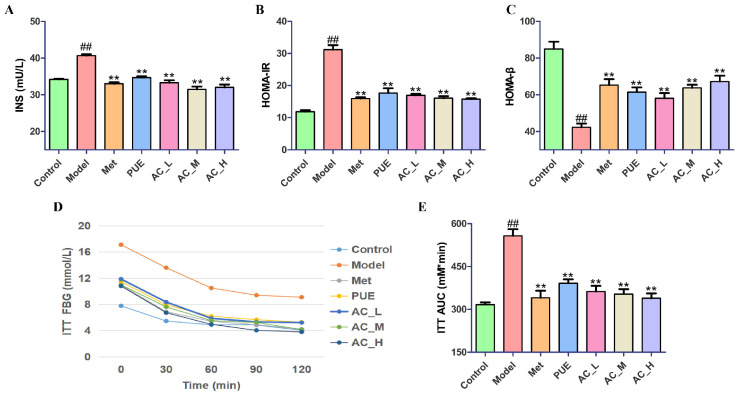
Influence of *P. lobata* on insulin sensitivity in STZ-induced T2DM mice. (**A**) Levels of insulin in serum; (**B**) HOMA-IR values; (**C**) HOMA-β values; (**D**) Insulin tolerance test (ITT); (**E**) AUC for ITT. In comparison to the control group, ^##^
*p* < 0.01; when contrasted with the model group, ** *p* < 0.01.

**Figure 9 pharmaceuticals-17-01675-f009:**
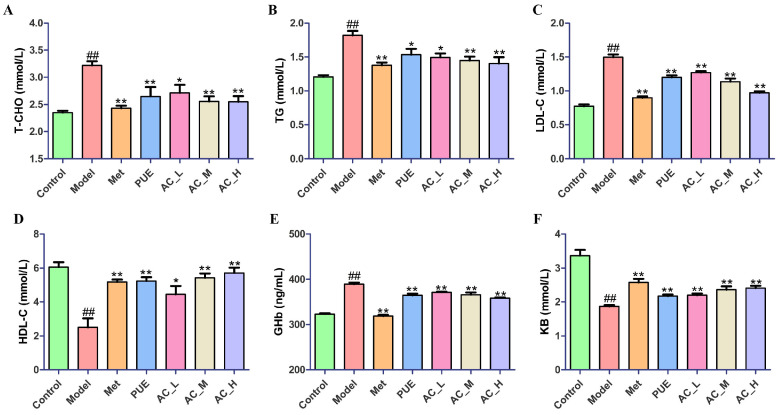
Improvement of *P. lobata* extract and active constituent on biochemical indicators in STZ-induced T2DM mice. Levels of T-CHO (**A**), TG (**B**), LDL-C (**C**), HDL-C (**D**), GHb (**E**), and KB (**F**) in serum. In comparison to the control group, ^##^ *p* < 0.01; when contrasted with the model group, * *p* < 0.05, ** *p* < 0.01.

**Figure 10 pharmaceuticals-17-01675-f010:**
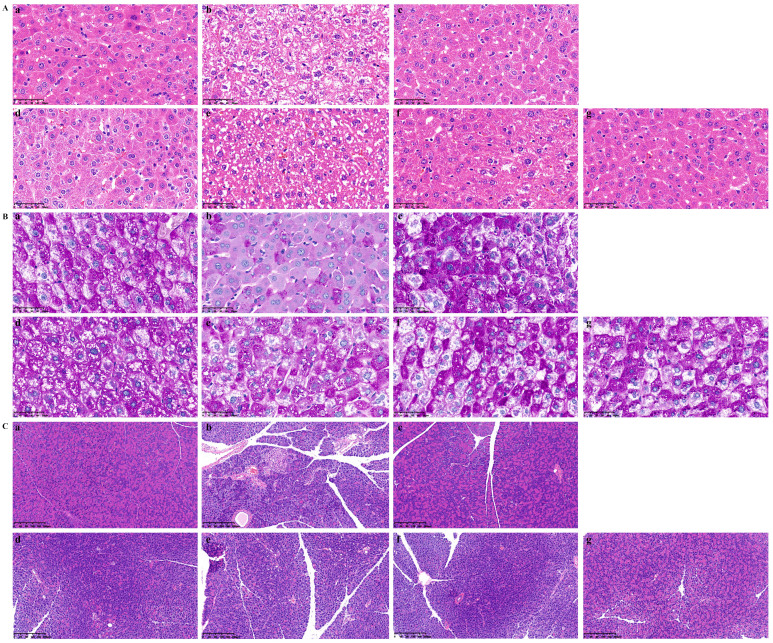
Histological evaluation of the effects of *P. lobata* extract and active constituent on liver and pancreas tissues in STZ-induced T2DM mice. In the results of H&E staining (**A**) and PAS staining (**B**) of liver (magnification, ×400), and H&E staining of pancreas (**C**) (magnification, ×100), a–g represent the Control group, Model group, Metformin (Met) group, PUE group, and AC groups (50, 100, 200 mg/kg).

**Table 1 pharmaceuticals-17-01675-t001:** MS data of 85 blood components (peaks) from rats.

No	Identified Components	M (*m*/*z*)	Formular	Error (ppm)	Adduct
1	Gallic acid+O-H_2_+SO_3_	262.952	C_7_H_4_O_9_S	6.6	−H
2	Gallic acid-H_2_+C_2_H_2_O	209.011	C_9_H_6_O_6_	8.4	−H
3	3′-Hydroxy-6″-O-xylosylpuerarin-C_21_H_18_O_10_(cleavage)-H_2_+C_2_H_2_O	173.045	C_7_H_10_O_5_	−5.5	−H
4	3-methoxy pyridine+H_2_+C_10_H_15_N_3_O_6_S	415.130	C_16_H_24_N_4_O_7_S	1.5	−H
5	4′,6,7-trihydroxyisoflavone-6-methylether-7-o-β-d-xylopyranosyl-(1→6)-β-d-glucopyranoside-C_16_H_10_O_4_(cleavage)+C_2_H_2_O	353.109	C_13_H_22_O_11_	0.3	−H
6	4′,6″-diacetyl puerarin-C_23_H_22_O_9_(cleavage)-H_2_+SO_3_	136.954	C_2_H_2_O_5_S	−9	−H
7	4-Hydroxy-3-methoxy cinnamic acid+2x(+H_2_)+C_2_H_2_O	239.090	C_12_H_16_O_5_	−9.9	−H
8	4-Hydroxy-3-methoxy cinnamic acid+H_2_+C_6_H_8_O_6_	371.099	C_16_H_20_O_10_	0.3	−H, +HCOO
9	4-Hydroxy-3-methoxy cinnamic acid+O	255.052	C_10_H_10_O_5_	3.8	+HCOO
10	4-Hydroxy-3-methoxy cinnamic acid+O+SO_3_	289.003	C_10_H_10_O_8_S	1.2	−H
11	4-Hydroxy-3-methoxy cinnamic acid-CH_2_(cleavage)+2x(+O)+C_2_H_2_O	253.034	C_11_H_10_O_7_	−5.2	−H
12	4-Hydroxy-3-methoxy cinnamic acid-CH_2_O(cleavage)+SO_3_	242.998	C_9_H_8_O_6_S	4.6	−H
13	4-Hydroxy-3-methoxy cinnamic acid-H_2_+SO_3_	317.000	C_10_H_8_O_7_S	7.5	+HCOO
14	6″-O-Acetyldaidzin-C_15_H_8_O_3_(cleavage)+2x(+H_2_)+SO_3_	351.058	C_8_H_18_O_10_S	−6.5	+HCOO
15	6″-O-Acetyldaidzin-C_15_H_8_O_3_(cleavage)+H_2_+C_2_H_2_O	265.093	C_10_H_18_O_8_	0	−H
16	6″-O-Acetyldaidzin-C_15_H_8_O_3_(cleavage)+O+H_2_+C_2_H_2_O	281.088	C_10_H_18_O_9_	0.6	−H
17	6″-O-Acetyldaidzin-C_15_H_8_O_4_(cleavage)+2x(+H_2_)+SO_3_	335.065	C_8_H_18_O_9_S	0.1	+HCOO
18	6″-O-malonyl genistin-C_15_H_8_O_4_(cleavage)+H_2_+SO_3_	347.028	C_9_H_16_O_12_S	−3.3	−H
19	6″-O-malonyl genistin-C_15_H_8_O_5_(cleavage)+H_2_-H_2_+C_2_H_2_O	337.078	C_11_H_16_O_9_	1.5	+HCOO
20	6″-O-malonyl genistin-C_15_H_8_O_5_(cleavage)-H_2_+C_2_H_2_O	335.059	C_11_H_14_O_9_	−7.6	+HCOO
21	6″-O-malonyldaidzin-C_15_H_8_O_3_(cleavage)+2x(+H_2_)+SO_3_	349.042	C_9_H_18_O_12_S	−6.8	−H
22	6″-O-malonyldaidzin-C_15_H_8_O_3_(cleavage)+2x(+O)+SO_3_	423.009	C_9_H_14_O_14_S	0	+HCOO
23	6″-O-malonyldaidzin-C_15_H_8_O_3_(cleavage)+2x(−H_2_)+SO_3_	340.982	C_9_H_10_O_12_S	1.2	−H
24	6″-O-malonyldaidzin-C_15_H_8_O_3_(cleavage)+H_2_+SO_3_	347.028	C_9_H_16_O_12_S	−3	−H
25	6″-O-malonyldaidzin-C_15_H_8_O_4_(cleavage)+2x(+H_2_)+SO_3_	333.050	C_9_H_18_O_11_S	−0.3	−H
26	6″-O-malonyldaidzin-C_21_H_18_O_8_(cleavage)+2x(−H_2_)+C_10_H_15_N_3_O_6_S	404.039	C_13_H_15_N_3_O_10_S	−2.7	−H
27	6″-O-malonyldaidzin-C_21_H_18_O_8_(cleavage)+C_10_H_15_N_3_O_6_S	408.070	C_13_H_19_N_3_O_10_S	−4.4	−H
28	4′,7-dihdroxy-3′-methoxyisoflavone 8-C-[β-D-glucopyranosyl-(1→6)]-β-D-glucopyranoside	608.174	C_28_H_32_O_15_	0.1	+HCOO
29	8-C-α-glucofuranosyl-7,3′,4′-trihydroxyisoflavone-O(cleavage)+2x(−H_2_)+C_2_H_2_O	451.102	C_24_H_20_O_9_	−3.8	−H
30	8-prenyldaidzein-O(cleavage)+C_2_H_2_O	347.131	C_22_H_20_O_4_	5.7	−H
31	8-prenyldaidzein-O(cleavage)+H_2_+C_2_H_2_O	349.145	C_22_H_22_O_4_	1.7	−H
32	Acetyl-kaikasaponin III-C_35_H_54_O_7_(cleavage)-H_4_	335.065	C_12_H_16_O_11_	9.7	−H
33	Acetyl-soyasaponin I-C_42_H_66_O_11_(cleavage)+O-H_2_+C_2_H_2_O	249.027	C_8_H_10_O_9_	8.8	−H
34	bis(2-ethylhexyl) phthalate-C_16_H_20_O_4_(cleavage)+H_2_+SO_3_	195.105	C_8_H_20_O_3_S	−7.1	−H
35	bis(2-ethylhexyl) phthalate-C_16_H_20_O_4_(cleavage)-H_2_+C_6_H_8_O_6_	333.154	C_14_H_24_O_6_	−4.6	+HCOO
36	bis(2-ethylhexyl) phthalate-C_8_H_16_O(cleavage)-H_2_+C_2_H_2_O	347.151	C_18_H_22_O_4_	1.8	+HCOO
37	Butesuperin A-C_16_H_8_O_3_(cleavage)+H_2_	215.091	C_10_H_16_O_5_	−8.2	−H
38	Corchoionoside C-C_6_H_10_O_5_(cleavage)	269.139	C_13_H_20_O_3_	0	+HCOO
39	Corylin-H_2_+SO_3_	397.035	C_20_H_14_O_7_S	−9.1	−H
40	crotonine+O+H_2_+SO_3_	393.062	C_12_H_16_N_2_O_8_S	2.8	+HCOO
41	crotonine-O(cleavage)+2x(+H_2_)	237.124	C_12_H_18_N_2_O_3_	−2.2	−H
42	crotonine-O(cleavage)+2x(−H_2_)+SO_3_	355.028	C_12_H_10_N_2_O_6_S	9.8	+HCOO
43	crotonine-O(cleavage)-H_2_	277.083	C_12_H_12_N_2_O_3_	0.4	+HCOO
44	Daidzein-4,7-diglucoside-C_6_H_10_O_5_(cleavage)+H_2_	498.141	C_21_H_25_O_11_	7.1	+HCOO
45	Diisobutyl phthalate+O+C_2_H_2_O	335.150	C_18_H_24_O_6_	−0.2	−H
46	Diisobutyl phthalate+O-H_2_+C_2_H_2_O	333.136	C_18_H_22_O_6_	6.4	−H
47	Diisobutyl phthalate-C_12_H_12_O_4_(cleavage)+2x(+H_2_)+SO_3_	187.066	C_4_H_14_O_3_S	9.6	+HCOO
48	Diisobutyl phthalate-C_4_H_8_(cleavage)+2x(+H_2_)+C_2_H_2_O	267.124	C_14_H_20_O_5_	−1	−H
49	Diisobutyl phthalate-C_4_H_8_(cleavage)+H_2_+SO_3_	303.056	C_12_H_16_O_7_S	5.8	−H
50	Diisobutyl phthalate-C_4_H_8_(cleavage)+O+C_2_H_2_O	279.089	C_14_H_16_O_6_	6.3	−H
51	Diisobutyl phthalate-C_4_H_8_(cleavage)-H_2_+C_2_H_2_O	307.082	C_14_H_14_O_5_	−0.5	+HCOO
52	Diisobutyl phthalate-C_4_H_8_O(cleavage)+C_2_H_2_O	293.103	C_14_H_16_O_4_	−0.4	+HCOO
53	Diisobutyl phthalate-C_4_H_8_O(cleavage)+H_2_	253.106	C_12_H_16_O_3_	−8.3	+HCOO
54	Formononetin-CH_2_O(cleavage)+2x(+H_2_)+SO_3_	321.044	C_15_H_14_O_6_S	−1	−H
55	hesperidin+2x(+O)+C_10_H_15_N_3_O_6_S	992.246	C_38_H_49_N_3_O_23_S	0.5	+HCOO
56	hesperidin-C_16_H_12_O_6_(cleavage)+2x(+H_2_)	359.156	C_12_H_26_O_9_	1.3	+HCOO
57	Irisolidone+O-H_2_+SO_3_	453.013	C_17_H_12_O_10_S	0.2	+HCOO
58	kakkasaponin I-C_36_H_56_O_7_(cleavage)	295.103	C_11_H_20_O_9_	−1.7	−H
59	kakkasaponin I-C_36_H_56_O_8_(cleavage)	279.107	C_11_H_20_O_8_	−5.1	−H
60	kudzusaponin A_1_-C_47_H_74_O_18_(cleavage)+2x(+H_2_)+SO_3_	249.028	C_5_H_14_O_9_S	−2.6	−H
61	kudzusaponin A_1_-C_47_H_74_O_18_(cleavage)+2x(-H_2_)+C_2_H_2_O	249.027	C_7_H_8_O_7_	7	+HCOO
62	kudzusaponin A_1_-C_47_H_74_O_18_(cleavage)+O+SO_3_	260.993	C_5_H_10_O_10_S	5.1	−H
63	kudzusaponin C_1_-C_48_H_78_O_16_(cleavage)	239.044	C_6_H_10_O_7_	13.4	+HCOO
64	kudzusaponin SA_1_-C_30_H_48_O_4_(cleavage)-H_4_	335.063	C_12_H_16_O_11_	1.8	−H
65	kudzusaponin SA_4_+2x(+O)+SO_3_	1115.420	C_47_H_74_O_25_S	−2.3	+HCOO
66	kudzusaponin SA_4_-C_35_H_56_O_7_(cleavage)	369.065	C_12_H_18_O_13_	−5.7	−H
67	kudzusaponin SA_4_-C_35_H_56_O_8_(cleavage)-H_2_	351.057	C_12_H_16_O_12_	0.5	−H
68	kudzusaponin SB_1_-C_12_H_21_O_9_(cleavage)+O+SO_3_	906.392	C_41_H_65_O_17_S	−0.8	+HCOO
69	Palmitic acid+2x(−H_2_)+SO_3_	331.158	C_16_H_28_O_5_S	−1.4	−H
70	puerarin	415.104	C_21_H_20_O_9_	0.3	−H
71	pueroside a+O+C_10_H_15_N_3_O_6_S	926.253	C_39_H_49_N_3_O_21_S	2.1	−H
72	pulobatones A+2x(+H_2_)+C_2_H_2_O	313.145	C_19_H_22_O_4_	−0.1	−H
73	pulobatones A+O+SO_3_	363.054	C_17_H_16_O_7_S	−0.1	−H
74	pulobatones B-C_16_H_8_O_3_(cleavage)+O+C_2_H_2_O	433.138	C_18_H_26_O_12_	5.7	−H
75	pulobatones B-C_16_H_8_O_4_(cleavage)-H_2_+C_2_H_2_O	445.136	C_18_H_24_O_10_	3	+HCOO
76	quercetin 3-O-β-D-glucofuranoside+2x(+O)	495.075	C_21_H_20_O_14_	−5.4	−H
77	sissotorin-C_6_H_10_O_5_(cleavage)+C_2_H_2_O	377.121	C_18_H_20_O_6_	−7.7	+HCOO
78	sissotorin-C_6_H_10_O_5_(cleavage)+O+H_2_+SO_3_	387.073	C_16_H_20_O_9_S	−6.7	−H
79	sissotorin-C_6_H_10_O_6_(cleavage)+2x(+H_2_)+C_2_H_2_O	365.158	C_18_H_24_O_5_	−8.1	+HCOO
80	sophoracoumestan A+O+C_2_H_2_O	391.081	C_22_H_16_O_7_	−2.5	−H
81	soyasapogenol A-H_4_	337.079	C_12_H_18_O_11_	5.1	−H
82	Sucrose-H_4_	337.080	C_12_H_18_O_11_	7.4	−H
83	Tectorigenin+O-H_2_+SO_3_	438.998	C_16_H_10_O_10_S	0.4	+HCOO
84	tuberosin+2x(+H_2_)	341.135	C_20_H_22_O_5_	0.9	−H
85	Vanillin+O+C_6_H_8_O_6_	343.067	C_14_H_16_O_10_	0.4	−H

**Table 2 pharmaceuticals-17-01675-t002:** Characteristic parameters of potential active components of *P. lobata* in the network.

Components	Degree	Betweenness	Closeness
Puerarin	90	0.53	0.43
Formononetin	67	0.35	0.41
Daidzein-4,7-diglucoside	63	0.19	0.40
4′,7-dihdroxy-3′-methoxyisoflavone	54	0.08	0.39
Tuberosin	17	0.04	0.35
Pulobatones A	13	0.01	0.35
6″-*O*-Acetyldaidzin	12	0.02	0.35
4′,6″-diacetyl puerarin	8	0.01	0.35
Kudzusaponin A1	5	0.00	0.34
Pulobatones B	4	0.00	0.34
Corchoionoside C	1	0.00	0.34
3′-Hydroxy-6″-*O*-xylosylpuerarin	1	0.00	0.34

## Data Availability

The study data are contained within the article.

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
