# Peer review of "Scientific Validation of Using Active Constituent as Research Focus in Traditional Chinese Medicine: Case Study of Pueraria lobata Intervention in Type 2 Diabetes"

_pharmaceuticals, 2024, doi:10.3390/ph17121675_

Round 1

Reviewer 1 Report

Comments and Suggestions for Authors

This is an interesting manuscript and extensive involving fractionation of P. lobata extracts to obtain biologically active components that affect aspects of Type 2 Diabetes.  The authors characterized 12 active components (out of 85 components identified from blood from rats injected with the extract) that were apparently effective.  They also identified a group of 270 target genes that were likely targets.  Using IR-HepG2 cells, they also identified 5 active compounds that affected glycogen metabolism in the cells.  Using mice, they evaluated effects of compounds, relative to metformin, on a diabetic model (STZ-treatment).

However, the authors should define abbreviations when first used; it is problematic when authors assume the reader knows all of these: for example, what is INS?

I also recommend that the authors use consistent nomenclature:  for example  Mix and active constituent. 

The Materials and Methods Section needs to be clarified and expanded:  

1.       Line 420:  was entire plant mashed or only selected parts?

2.       Line 417 : add (PUE) at the end of this sentence; also how were filtrates evaporated?

3.       The use of 36 SD male rats divided into 6 groups is very unclear; one group is control, 1 group is model, one group is met addition, one group is PUE addition but what are the other 2 groups?  This is very unclear.  Also, were they given STZ?  For line 439:  what was administered to the rats by gavage for 4 weeks? 

4.       Line 485 makes no sense

5.       Line 557: where was trypsin obtained from?

6.       Line 571:  what is AG group?  If used 3 different MIX (50, 100, 200 mg/kg/day) do you have 6 or 7 total groups?

7.       Line 587: how was blood obtained?

8.       Line 593:  what type of glucose meter was used

9.       Section 4.6.9:  when were samples obtained?  Reference for how they were processed?

Also the authors give no rationale for why both rats and mice were used in different parts of the study.  They should also clearly indicate in figure legends when rats or mice are used in the experiment. 

For Table 1, indicate at what time these data were obtained

For Figure 4 label the column to the right (for example: kJ/mol)

For line 172:is this the correct unit for the INS solution?

For Figure 7:  where are the L, M, H mixes defined?

Line 256: add…..in some parameters but not all (3 of 6 were significantly different for mix L)

Lines 322-325: these are not the same names as in Figure 4B

The authors have not commented on the efficacy of the PUE or individual components or Mix relative to the results using metformin. 

Comments on the Quality of English Language

The authors have a number of sentences that are very difficult to read and understand; for example line 485.

Author Response

This is an interesting manuscript and extensive involving fractionation of P. lobata extracts to obtain biologically active components that affect aspects of Type 2 Diabetes. The authors characterized 12 active components (out of 85 components identified from blood from rats injected with the extract) that were apparently effective. They also identified a group of 270 target genes that were likely targets. Using IR-HepG2 cells, they also identified 5 active compounds that affected glycogen metabolism in the cells. Using mice, they evaluated effects of compounds, relative to metformin, on a diabetic model (STZ-treatment).

Dear Reviewer, we sincerely appreciate your thorough review and constructive suggestions regarding our manuscript. We have revised the manuscript in accordance with your comments and have marked the amendments in the revised version. Below are our point-by-point responses to the comments:

However, the authors should define abbreviations when first used; it is problematic when authors assume the reader knows all of these: for example, what is INS?

Definitions for abbreviations that appear in the manuscript, such as INS, OGTT, ITT, PUE, etc. have been added upon their first occurrence.

I also recommend that the authors use consistent nomenclature: for example Mix and active constituent.

Following your suggestion, 'Mix' and 'active constituent' have been consistently renamed as 'active constituent' and abbreviated as 'AC'. Additionally, corresponding modifications have been made in the figures.

The Materials and Methods Section needs to be clarified and expanded:

  1. Line 420: was entire plant mashed or only selected parts?

The root of Pueraria lobata (Willd.) Ohwi was used in this research, and the description has been rewritten as ‘The root of P. lobata (1 kg) were mashed’.

  1. Line 417: add (PUE) at the end of this sentence; also how were filtrates evaporated?

‘(PUE)’ has been added at the end of this sentence. In the experiment, a vacuum rotary evaporator was used to concentrate the filtrate, which has been supplemented in the manuscript.

  1. The use of 36 SD male rats divided into 6 groups is very unclear; one group is control, 1 group is model, one group is met addition, one group is PUE addition but what are the other 2 groups? This is very unclear. Also, were they given STZ? For line 439: what was administered to the rats by gavage for 4 weeks?

We sincerely apologize for the confusion caused by our previous incorrect description. In this section, we utilized 24 SD rats, which were divided into four groups: the control group, model group, Met group, and PUE group. The model group, Met group, and PUE group received intraperitoneal injections of STZ, while the control group was administered an equal volume of buffer solution. Following the successful establishment of the T2DM model, the rats were subjected to gavage for 4 weeks. During this period, the control group and the model group were given water, the Met group received metformin, and the PUE group was administered P. lobata extract. We have revised the description of this section in the manuscript accordingly.

  1. Line 485 makes no sense

This sentence appeared in an inappropriate place and has been deleted.

  1. Line 557: where was trypsin obtained from?

The trypsin used in the cell experiment is 0.25% trypsin-EDTA from Gibco, USA. This information has been incorporated into the manuscript.

  1. Line 571: what is AG group? If used 3 different MIX (50, 100, 200 mg/kg/day) do you have 6 or 7 total groups?

We sincerely apologize for the confusion caused by our incorrect description. In this section, we utilized 42 mice, which were divided into 7 groups: the control group, model group, Met group, PUE group and AC groups (50, 100, 200 mg/kg/day). Additionally, the term 'AG group' is a writing error; it should be corrected to 'Met group,' which serves as the positive control group.

  1. Line 587: how was blood obtained?

Blood to measure fasting glucose was obtained from the tail vein of mice. This information has been incorporated into the manuscript.

  1. Line 593: what type of glucose meter was used

The glucose meter is ACCU-CHEK Active Blood Glucose Meter obtained from F. Hoffmann-La Roche Ltd, Germany. This information has been incorporated into the manuscript.

  1. Section 4.6.9: when were samples obtained? Reference for how they were processed?

The sample acquisition time and method are added in Section 4.6.5. Specifically, following a 10-week period of gavage, each group of mice underwent an 8-hour fasting period before blood samples were retrieved from the orbital vein. Afterward, the mice were euthanized via the cervical dislocation technique, and their livers and pancreases were harvested for additional research.

The sample processing method is added in section 4.6.9, as follows: After gathering samples, small pieces of mouse liver tissue and the pancreas were preserved in 4% paraformaldehyde for 24 hours, followed by embedding and sectioning [34].

Also the authors give no rationale for why both rats and mice were used in different parts of the study. They should also clearly indicate in figure legends when rats or mice are used in the experiment.

We sincerely appreciate your thorough review and constructive suggestions regarding our manuscript. In response to your inquiries, we provide the following clarification: The blood concentration of components in oral Chinese medicine is typically low, and certain constituents in drug-containing serum may fall below the detection limit, which complicates accurate identification during analysis. However, sufficient quantities of drug-containing serum can be obtained from rats, enabling a more comprehensive mass spectrometry analysis to examine the blood components of P. lobata through component enrichment. Consequently, we employed rats as experimental subjects in the spectrum-effect correlation analysis. In accordance with your suggestion, we have added description of the type of animals used in the titles and legends of the relevant charts.

For Table 1, indicate at what time these data were obtained

After four weeks of oral administration, the rats in each group were fasted overnight, and blood was collected from the abdominal aorta following anesthesia. The serum was then divided into two portions: one was utilized for mass spectrometry analysis to identify the blood components of P. lobata, while the other was reserved for the assessment of serum indicators. The relevant description is supplemented in section 4.3.2.

For Figure 4 label the column to the right (for example: kJ/mol)

Figure 4A is redrawn with labels placed on the right.

For line 172: is this the correct unit for the INS solution?

The standard unit of insulin solution is expressed as ‘U/mL’ or ‘IU/mL’. In cellular experiments, for clarity, unit conversion was performed using the following formula: 1 IU/mL = 0.001667 mM. We have now reverted it back to ‘IU/mL’.

For Figure 7: where are the L, M, H mixes defined?

We defined AC_L, AC_M, and AC_H and added them in section 4.6.5.

Line 256: add…..in some parameters but not all (3 of 6 were significantly different for mix L)

It has been added in the appropriate location according to your suggestion.

Lines 322-325: these are not the same names as in Figure 4B

We have replaced the docking results in Figure 4B. Thank you very much for your careful review, which has made our manuscript more rigorous.

The authors have not commented on the efficacy of the PUE or individual components or Mix relative to the results using metformin.

We have added some comments on the effects of active constituent and metformin on T2DM in the Results and Discussion sections.

The authors have a number of sentences that are very difficult to read and understand; for example line 485.

We sincerely apologize for any inconvenience caused by the inaccuracies in our expressions. In response to your comment to the quality of the English language, we have carefully proofread the manuscript to correct any minor errors. We endeavour to ensure that the language is clearer and more precise. Thank you again for your recognition and suggestions.

Reviewer 2 Report

Comments and Suggestions for Authors

The manuscript introduces a novel framework of studying TCM by focusing on active constituents rather than crude extracts. This methodology bridges traditional practices with modern scientific methods. The study uses Pueraria lobata as a model to explore synergistic effects, providing a new avenue for TCM research. The inclusion of spectrum-effect correlation analysis, network pharmacology, molecular docking, and both in vivo and in vitro studies enhances the reliability and robustness of the results.

The results are well-supported with tables, figures, and statistical analysis, making the study easy to interpret. The stepwise identification of five active components highlights a systematic research process.

Taken together, the manuscript provides a solid contribution to the field of TCM and pharmacological research, offering a novel approach to studying multi-component therapies. Addressing the weaknesses would significantly enhance the impact and applicability of this work.

1.While molecular docking provides initial insights into potential interactions, the manuscript lacks in-depth discussion on how these interactions translate into observed physiological effects. Elucidating mechanisms such as how these compounds influence insulin sensitivity or β-cell repair could strengthen the study.

2.The absence of discussion on long-term safety and potential side effects leaves gaps in understanding practical applications.

3.There is no explicit mention of ethical approvals for animal studies, which could raise concerns about compliance with ethical standards. Including these details is crucial for credibility.

4.While the study identifies active components and demonstrates their effects, the mechanisms underlying these effects are not fully elaborated. For example, how do the identified active constituents modulate insulin sensitivity at the molecular level?

5.The manuscript would benefit from professional editing to improve flow and readability, as some sections are verbose or complex.

Author Response

The manuscript introduces a novel framework of studying TCM by focusing on active constituents rather than crude extracts. This methodology bridges traditional practices with modern scientific methods. The study uses Pueraria lobata as a model to explore synergistic effects, providing a new avenue for TCM research. The inclusion of spectrum-effect correlation analysis, network pharmacology, molecular docking, and both in vivo and in vitro studies enhances the reliability and robustness of the results.

The results are well-supported with tables, figures, and statistical analysis, making the study easy to interpret. The stepwise identification of five active components highlights a systematic research process.

Taken together, the manuscript provides a solid contribution to the field of TCM and pharmacological research, offering a novel approach to studying multi-component therapies. Addressing the weaknesses would significantly enhance the impact and applicability of this work.

Dear Reviewer, we sincerely appreciate your thorough review and constructive suggestions regarding our manuscript. We have revised the manuscript in accordance with your comments and have marked the amendments in the revised version. Below are our point-by-point responses to the comments:

1. While molecular docking provides initial insights into potential interactions, the manuscript lacks in-depth discussion on how these interactions translate into observed physiological effects. Elucidating mechanisms such as how these compounds influence insulin sensitivity or β-cell repair could strengthen the study.

Response: We sincerely appreciate your support and review of our manuscript. Your suggestions are highly constructive. As you indicated, conducting more in-depth research on the mechanisms will enhance our understanding of the scientific validity and feasibility of utilizing active constituents as the focal point for investigating the efficacy of Traditional Chinese Medicine (TCM). This will also guide the direction of our future research endeavors. Once again, thank you for your recognition and invaluable suggestions.

2. The absence of discussion on long-term safety and potential side effects leaves gaps in understanding practical applications.

Response: We sincerely appreciate your support and review of our manuscript. Your suggestions are invaluable. However, since Pueraria lobata is a traditional Chinese medicine with homology of medicine and food, it is considered to have no toxic effects or side effects. Consequently, we have not discussed its long-term safety or potential side effects at this stage. In the subsequent phase of our research on the preparation, we will ensure that the safety of these preparations is thoroughly evaluated.

3. There is no explicit mention of ethical approvals for animal studies, which could raise concerns about compliance with ethical standards. Including these details is crucial for credibility.

Response: We sincerely appreciate your thorough review and constructive suggestions regarding our manuscript. Following your suggestion, the ethical approval for animal studies has been added in the animal model section and marked in red.

4. While the study identifies active components and demonstrates their effects, the mechanisms underlying these effects are not fully elaborated. For example, how do the identified active constituents modulate insulin sensitivity at the molecular level?

Response: We sincerely appreciate your valuable advice and recognize the importance of your question. We regret that our current data do not offer additional insight into the regulatory mechanisms by which the active constituent of Pueraria lobata influence insulin sensitivity. In the future, we will prioritize improving our experimental design to address this gap.

5. The manuscript would benefit from professional editing to improve flow and readability, as some sections are verbose or complex.

Response: We appreciate your attention to the linguistic aspects of our manuscript. We understand that clear and concise communication is essential for scientific writing. In response, we have carefully revised the language, paying attention to sentence structure, vocabulary choices, and tone. We believe that these improvements will enhance the accessibility and engagement of our paper for readers.

Round 2

Reviewer 1 Report

Comments and Suggestions for Authors

I thank the authors for their appropriate and useful revisions...nicely done.

Line 187 is not clearly written as a 'good' sentence. It still needs to be revised.

Author Response

I thank the authors for their appropriate and useful revisions ... nicely done.

Line 187 is not clearly written as a 'good' sentence. It still needs to be revised.

Response: Dear Reviewer, We sincerely appreciate your thorough review of our manuscript and for recognizing our work. In response to your suggestion, we have revised the sentence on line 187 for improved clarity. The rewritten sentence is as follows: "Subsequently, the glucose consumption and glycogen levels of IR-HepG2 cells were assessed at the effect concentrations of the 5 active components, PUE, and AC." We hope that this modification meets your requirements. Thank you once again for your recognition and valuable suggestions.